# The Effect of Conductive Additive Morphology and Crystallinity on the Electrochemical Performance of Ni-Rich Cathodes for Sulfide All-Solid-State Lithium-Ion Batteries

**DOI:** 10.3390/nano13233065

**Published:** 2023-12-01

**Authors:** Jae Hong Choi, Sumyeong Choi, Tom James Embleton, Kyungmok Ko, Kashif Saleem Saqib, Jahanzaib Ali, Mina Jo, Junhyeok Hwang, Sungwoo Park, Minhu Kim, Mingi Hwang, Heesoo Lim, Pilgun Oh

**Affiliations:** 1Department of Smart Green Technology Engineering, Pukyong National University, 45, Yongso-ro, Nam-gu, Busan 48547, Republic of Korea; critical316@naver.com (J.H.C.); csmkie9805@gmail.com (S.C.); tembleton77@gmail.com (T.J.E.); ahrl7237@gmail.com (K.K.); kashifsaqib90@gmail.com (K.S.S.); jahanzaib.ali12@yahoo.com (J.A.); alskgmadl98@gmail.com (M.J.); junhyeock74@gmail.com (J.H.); tjddn8117@gmail.com (S.P.); skkim633@gmail.com (M.K.); t66549@gmail.com (M.H.); gmltn6649@naver.com (H.L.); 2Department of Nanotechnology Engineering, Pukyong National University, 45, Yongso-ro, Nam-gu, Busan 48547, Republic of Korea

**Keywords:** conductive additive, morphology, all-solid-state lithium-ion batteries, carbon nanofiber

## Abstract

Sulfide electrolyte all-solid-state lithium-ion batteries (ASSLBs) that have inherently nonflammable properties have improved greatly over the past decade. However, determining both the stable and functional electrode components to pair with these solid electrolytes requires significant investigation. Solid electrolyte comprises 20–40% of the composite cathode electrode, which improves the ionic conductivity. However, this results in thick electrolyte that blocks the electron pathways in the electrode, significantly lowering the electrochemical performance. The application of conductive carbon material is required to overcome this issue, and, hence, determining the carbon properties that result in the most stable performance in the sulfide solid electrolyte is vital. This study analyzes the effect of the cathode conductive additive’s morphology on the electrochemical performance of sulfide electrolyte-based ASSLBs. Carbon black (CB) and carbon nanotubes (CNTs), which provide electron pathways at the nanoscale and sub-micron scale, and carbon nanofiber (CNF), which provides electron pathways at the tens-of-microns scale, are all tested individually as potential conductive additives. When the CNF, with its high crystallinity, is used as a conductive material, the electrochemical performance shows an excellent initial discharge capacity of 191.78 mAh/g and a 50-cycle capacity retention of 83.9%. Conversely, the CB and the CNTs, with their shorter pathways and significantly increased surface area, show a relatively low electrochemical performance. By using the CNF to provide excellent electrical conductivity to the electrode, the polarization is suppressed. Furthermore, the interfacial impedance across the charge transfer region is also reduced over 50 cycles compared with the CB and CNT composite cells. These findings stringently analyze and emphasize the importance of the morphology of the carbon conductive additives in the ASSLB cathode electrodes, with improvements in the electrochemical performance being realized through the application of long-form two-dimensional crystalline CNFs.

## 1. Introduction

Lithium-ion batteries (LIBs) are currently considered to be one of the primary solutions to the energy storage problem due to their unique advantages of being lightweight and having a high specific energy density, high capacity, high efficiency, and a long cycle life [1,2]. Due to the recent uptick in the popularity of electric vehicles and energy storage devices, the market for LIBs has grown significantly over the past few decades following the diversification of battery applications [3]. However, the development of liquid electrolyte-based lithium-ion batteries is limited by several component properties, such as the flammability of the standard organic electrolyte material [4,5], the growth of lithium dendrites [6], and the side reactions of the electrolyte affecting the reversible capacity and the thermal stability [7,8]. In order to avoid the dangers of liquid electrolytes, which can lead to combustion [9], research on all-solid-state lithium-ion batteries (ASSLBs) that have inherently nonflammable, inorganic electrolytes has garnered significant attention [10,11]. Research into ASSLBs has been extensively reported since the development of sulfide solid electrolytes (SEs) with ionic conductivity values of ~2.5 × 10^−2^ S·cm^−1^, which are higher than those of certain liquid electrolytes [12,13]. Out of these sulfide SEs, Li_6_PS_5_X (with X = Cl^−^, Br^−^, I^−^) argyrodite-type candidates have been considered as some of the most practically viable SEs, as they are relatively high in ionic conductivity and are somewhat stable compared to Lithium metal anodes [14]. However, despite the development of SEs with high ionic conductivity, multiple challenges remain to be solved before the practical application of ASSLBs can be achieved. Currently, the electrochemical performance and power density of ASSLBs have not yet met commercial standards and have been unable to rival the performance of LIBs [8,15]. The most significant limitation of ASSLBs is that the solid nature of the SEs results in poor contact with the active material, and, hence, the ionic conductivity in the electrode is low. To overcome this, 20–40% of the electrode comprises solid electrolyte, which is mixed with the active material to increase the contact between the two [11]. This often leaves ASSLBs’ cells with an energy density <300 Wh kg^−1^ when practically evaluated at nominal voltage ~3.7 V, despite one of their key advantages being that their theoretical energy densities can be far higher at >400 Wh kg^−1^ [16]. Additionally, although the ionic conductivity is improved by adding the solid electrolyte, the electron pathways are blocked by the solid electrolyte resulting in poor electrical networks and electron-isolated active material [11].

Various cathodes have been researched and commercialized for application in LIBs, including Lithium Iron Phosphate (LFP), Lithium Cobalt Oxide (LCO), various Manganese Oxide materials (e.g., LMO), and various Ni-rich materials (LiNi_1-x_M_x_O_2_, M = Co, Mn) [17,18,19,20,21]. Due to their high capacity and low material cost, Ni-rich layered compounds have rapidly grown in popularity over the last decade [22,23]. Ni-rich cathode materials are very attractive for ASSLBs due to such advantages [24]. However, they do not significantly contribute to the electrical network of the cathode composite, as their electronic conductivity is very low [25]. Furthermore, they are particularly reactive and unstable, resulting in an abundance of research being conducted to overcome their many structural and chemical instabilities [26,27,28,29,30,31]. Hence, in typical LIB systems, the surface of the cathode material is coated with metal oxides, such as Al_2_O_3_ [32], Li_2_ZrO_3_ [33], MgO [34], and ZnO [35], to suppress the side reaction with the electrolyte and increase the stability of the cathode material. This concurrently increases the surface resistance of the cathode-active material, as the metal oxide coating is electrically non-conductive [36]. The poor electron conductivity of the overall electrode, including both the SE- and Ni-rich-active material, results in large overpotentials, especially at a high current density and high loading density [37]. Due to this lack of electrical conductivity in the active material and the SE, the electrochemical performance of the cathode is substantially hindered through the electrical isolation of the Ni-rich particles. Hence, carbon additives with high electrical conductivity are applied as conducting agents [38,39] and can form an electrical network between the active materials [40] to compensate for the naturally low electrical conductivity of the electrode. In particular, carbon black (CB), which is inexpensive and has excellent electrical conductivity and stable electrochemical properties, has been long considered as the standard carbon additive applied in LIBs. However, applying this point contact conductive additive in ASSLBs results in a weak conductive network that cannot easily penetrate the thick electrolyte. Furthermore, reactions between CB and SE have been previously determined as electrochemically inhibiting [41]. Uncovering the requirements of SE-based cathode composite carbon additives is crucial for overcoming the issues associated with the electrochemical performance of ASSLBs. Recently, studies using carbon nanotubes (CNTs) [38], graphene [42], carbon nanofiber (CNF) [39], and 3D carbon frames [43] with high electrical conductivity properties have been reported, in an attempt to reduce the content of conductive materials in electrodes due to the demand for a high energy density. However, work focusing purely on SE-based ASSLB systems is few and far between and is certainly disproportionate to the research interest that these systems are garnering.

In this work, we propose a strategy to increase the electrochemical performance of ASSLBs, focusing on the morphology and crystallinity of conductive additives using Ni-rich cathode materials (LiNi_0.8_Co_0.1_Mn_0.1_O_2_). Three types of commercial carbon, CB, CNTs, and CNF, are compared individually as conductive additives in the cathode electrode of the ASSLBs. Carbon black and CNTs are nano-sized materials, so sub-micron electron paths are formed through the interconnection of individual particles. On the other hand, the size of the CNF was several tens of microns, so the CNF provided electron pathways of various two-digit micron values in length. When these carbon materials were used as conductive additives, the composite containing the CNF showed excellent cycle stability (>83.9% capacity retention after 50 cycles at 0.5 C) and an excellent rate performance compared to the CB and CNT composites. Furthermore, the CNF and CNT composites, consisting of crystallized carbon, always showed higher coulombic efficiency throughout electrochemical evaluation. The CB composite, which consists of low-crystallization carbon/amorphous carbon, was poorer by comparison. By using the CNF to provide excellent electrical conductivity to the electrode, the polarization is suppressed and the interfacial impedance across the charge transfer region is also reduced across the cycles. Additionally, the excellent cycle ability is evidenced by the CNF stably maintaining a reversible H1-M phase transition. The analysis provided within this study will help to improve the standard heuristic for designing ASSLB cathode electrodes, with more consideration of the carbon additive likely to be taken as a direct result of these findings. The physical properties of each of the conductive materials were analyzed through Raman Spectroscopy, scanning electron microscopy (SEM), and powder X-ray diffraction (XRD), and the electrode impedance associated with each conductive material was analyzed through electrochemical impedance spectroscopy (EIS).

## 2. Experimental Section

### 2.1. Synthesis of NCM811

In accordance with the molar ratio of 8:1:1(Ni:Co:Mn transition metal ions), nickel sulfate (NiSO_4_·6H_2_O), cobalt sulfate (CoSO_4_·7H_2_O), and manganese sulfate (MnSO_4_·2H_2_O) were weighed and added to deionized water before being stirred, resulting in a homogenous solution with an overall concentration of metal ions of 0.56 mol/L. The 0.56 mol/L solution of metal ions suspended in water was placed in a continuously stirring tank reactor (CSTR) for the coprecipitation reaction. Amounts of 1 mol/L NaOH solution and 0.5 mol/L ammonia solution were added continuously throughout the stirring, acting as the precipitator and a complexing buffer, respectively. The entire reaction was continued for 24 h in an inert N_2_ atmosphere to entirely remove any contact with airborne contaminants. Throughout the reaction, the pH value of the coprecipitation solution was maintained between 10.8 to 11.4, and the temperature of the reactor was held at 50 °C. The coprecipitated particles were then separated from the solution via filtration. The particles were then washed and dried at 120 °C under vacuum conditions for 24 h to obtain the product Ni_0.8_Co_0.1_Mn_0.1_(OH)_2_ precursor. LiOH·H_2_O was then added to the precursor at a molar ratio of 1:1.03 before being mixed and heated at 500 °C for 5 h. Lastly, the mixed and heated product was calcined at 800 °C for 15 h, obtaining the Ni-rich layered oxide LiNi_0.8_Co_0.1_Mn_0.1_O_2_.

### 2.2. Materials Characterization

Carbon black (Timcal graphite and carbon Corp. Super C65, Bodio, Switzerland), multi-walled carbon nanotubes (CNano Corp. FT9110, (Zhenjiang, China),), and carbon nanofibers (E-cube materials (Jeonju, Republic of Korea)) were used as conductive additives. The crystalline structures of all the samples were determined through X-ray diffraction (D/Max2200, Rigaku, Tokyo, Japan) using Cu Kα radiation in the 2θ range of 10–80° with a step of angle 0.02°. Field emission scanning electron microscopy (FE-SEM, TESCAN, MIRA 3 LMH In-Beam Detector, Brno, Czech Republic) was performed to observe the particle surface morphology of the conductive materials with an applied voltage of 3 kV in back-scattered electron (BSE) mode. An SP-240 electrochemical workstation was utilized to obtain the electrochemical impedance spectra (EIS) in the frequency range from 10^5^ Hz to 0.1 Hz with an amplitude of 10 mV. Raman spectroscopy (JASCO, NRS-5100, Tokyo, Japan) is a well-known sensitive tool used to study the structural properties of carbonaceous materials. Raman was applied using a 532 nm laser with 3.2 mW. This was used to determine the graphitization of the carbon material applied. 

### 2.3. ASSLBs’ Assembly and Electrochemical Measurements

The ASSLBs’ cells were assembled with a coin cell architecture of 13 mm in diameter. The pre-prepared NCM811 powder was mixed with the Li_6_PS_5_Cl argyrodite solid electrolyte and the conductive material in a weight ratio of 66.6:28.6:4.8 (AM:SE:CM) as the composite cathode using an agate mortar and pestle. As a counter electrode, the Li and In were mixed in a ratio of 3:97, and then the SE was additionally mixed in a ratio of 4:1 with the Li-In composite. The ASSLBs’ cells were fabricated as follows. An amount of 150 mg of Li_6_PS_5_Cl solid electrolyte was pressed at 10 MPa with a 13 mm diameter to form the SE pellet in the center of the stack. An amount of 15 mg of composite cathode powder was uniformly spread on the positive side of the electrolyte pellet and was uniformly distributed across the surface. The Li-In powder was placed on the negative side of the electrolyte pellet and covered with Ni foil, acting as both a current collector and a layer of protection for the cell stack. Finally, all the components were compressed together at 50 MPa. All cell assemblies were performed in an Ar-filled glove box with H_2_O and O_2_, below 0.03 ppm to ensure the integrity of the data. The cells were charged and discharged between 1.88 and 3.68 (V vs. Li-In) at a current density of 0.1–0.5 C (1 C = 180 mA·g^−1^) and at a temperature of 30 °C with a WBCS 3000Le32 battery cycler from Won A Tech (Daejeon, Republic of Korea).

## 3. Results and Discussion

The size of the cathode material is several microns, and, therefore, in the case of the CB, which is a conductive material with a size of several tens to hundreds of nanometers, interconnections of multiple CB particles must be formed to create an electron path between the various active material particles (Figure 1a). In the case of the CNTs, the diameter is several nanometers, but the length varies from tens of nanometers to microns. However, most CNTs exist in a crumpled and entangled form, and the length of the electron path provided by CNTs is tens to hundreds of nanometers, similar to that of the CB. On the other hand, since the diameter of the CNFs is several microns and the length is several tens of microns, several cathode materials touching the surface of one CNF can directly receive the electrons. These conductive materials were mechanically mixed with the active material and the morphology of the composite mix was analyzed via the SEM. Figure 1b shows the morphology of the CB composite, where the CB is distributed around the active material evenly. Since the size of the CB is several tens of times smaller than that of the cathode material, several CB particles form interconnections to connect the active material particles. In the case of the CNTs, interestingly, most of the CNTs are attached on the surface of the cathode active material (Figure 1c) after mechanical mixing, unlike the CB. However, like the CB, the CNTs form a short electron path in the electrode that is only capable of electrically connecting active material particles in the direct vicinity. Therefore, it is only possible to form an electron pathway between the active material particles through connections via several conductive materials. On the other hand, in the case of the CNFs (Figure 1d), several active materials are in contact with one CNF. This means that even far distant active materials can form an electron path directly through one CNF. Higher-magnification SEM images, which clearly display the carbon morphologies, are supplied in Appendix A. Through analyzing the crystallinity of these carbon materials via XRD analysis (Figure 1e), the CB has a full width at half maximum (FWHM) of 5.73 degrees, the CNTs have a FWHM of 2.44 degrees and the CNF has a FW of 3.13 degrees. According to the Scherrer equation, the average crystal size of carbon materials increases as the FWHM of the (002) peak narrows [44]. So, the CNTs and CNFs can be considered as carbon materials with a higher crystallinity than the CB. In addition, in the Raman analysis (Figure 1f), the CNTs and CNFs have a strong G-peak (1580 cm^−1^) and 2D-peak (2700 cm^−1^), which are trademark footprints of a crystalline carbon form (Table 1). On the other hand, there is no G-peak and 2D-peak for the CB, which further confirms that it is an amorphous carbon. The CNTs and CNFs also show strong disorder peaks (D-peaks), which may be caused by the incomplete crystallization of carbon during carbonization [45].

The electrochemical analysis of the LiNi_0.8_Co_0.1_Mn_0.1_O_2_ (NCM) cathode composites with CB, CNTs and CNF as conductive additives was evaluated in ASSLBs to compare the initial discharge capacity and cycle properties. As shown in Figure 2a, when CB, CNTs and CNF were added as conductive additives in the composite alongside the SE and NCM, the NCM-CB, NCM-CNTs, and NCM-CNF composites exhibited charge capacities of 215.77, 240.00, and 223.04 mAh/g, discharge capacities of 152.01, 186.18, and 191.78 mAh/g, and initial coulombic efficiencies (ICE) of 70.45%, 77.58%, and 85.98%, respectively, through an evaluation of the ASSLBs within the potential range of 1.88–3.68 (V vs. Li-In) at 0.05 C (Table 2). In the case of the NCM-CB, the overpotential at the initial charging is higher than that of the NCM-CNTs and NCM-CNF. It also exhibited a relatively high overpotential throughout charging, and thus a relatively small amount of total charge capacity. Also, when charging, the overvoltage was relatively high when compared to the other samples, with the initial charge beginning at ~0.09 V vs. Li–In, which was greater than the other two samples that had comparable potentials. In the case of the NCM-CNTs, approximately 12 mAh/g of lithium was lost to side reactions during initial charging, as seen through the initial charge slope. The charge capacity attained was 240 mAh/g, which was the highest and is believed to be due to side reactions between the SE and CNTs attached on the surface of the active material, which showed a large adsorption surface area. On the other hand, in the case of the NCM-CNF, the slope of the initial charge was steep, signifying low side reactions. This was combined with a relatively low overpotential, resulting in what was the highest discharge capacity and ICE among the three samples. The NCM-CNF also shows a higher discharge capacity after the 1st cycle, achieving 163.6 mAh/g and a 50th cycle retention of 83.9%, compared to the NCM-CB (111.8 mAh/g of 1st discharge, 72.6% of 50th retention) and NCM-CNTs (137.8 mAh/g of 1st discharge, 67.4% of 50th retention) during cycling evaluation at 0.5 C (Figure 2b). In terms of the Columbic efficiency (Figure 2c), the NCM-CNF exhibited a higher value than the NCM-CB and NCM-CNTs on average. A comparison of this work’s electrochemical performance data and that of previous published works with similar chemistries is outlined in Appendix A.

Figure 3a displays the rate performance of the NCM-CB, NCM-CNTs and NCM-CNF at various discharge rates within the potential range of 1.88–3.68 (V vs. Li–In). During the rate evaluation, the charge rate was 0.05 C. The NCM-CNF shows a higher discharge capacity than the NCM-CB and NCM-CNTs at the 0.1 C, 0.2 C, 0.5 C, and 1 C rate evaluation. Moreover, it is seen from the charge–discharge curves of the NCM-CNF at different current rates in Figure 3b–e that the polarization of the NCM-CNF cathode is suppressed compared to that of the NCM-CB and NCM-CNTs, especially at high current rates where the polarization is significantly exaggerated. This indicates that the high electrical conductivity of the CNF in the electrode increases the power density and alleviates the polarization [36]. This, in effect, allows for an increase in the power density without significantly compromising the cyclability of the cell.

To analyze the effect of the conductive additives on the impedance between the cathode electrode and the solid electrolyte during the charge–discharge cycles, the electrochemical impedance spectroscopy (EIS) tests of the NCM-CB, NCM-CNTs and NCM-CNF were conducted after formation and after 50 cycles, as shown in Figure 4a,b, respectively. The EIS plots consist of one semicircle at high frequencies, one semicircle at middle frequencies, and a straight line at low frequencies. The semicircles at high and middle frequencies are related to the grain boundary impedance of the SE and to the interfacial charge transfer impedance between the cathode and SE, respectively [46]. The straight line is the Li^+^ diffusion impedance through the cathode electrode (Warburg diffusion (W)), and the intercept at the Z_re_ axis in the high frequency corresponds to the bulk impedance of the solid electrolyte [47,48]. The fitting data were evaluated according to these impedances, and the corresponding equivalent circuit to reflect this is provided in Appendix A. It is well known that the interface impedance between the cathode and solid electrolyte is the major contributor to the total cell impedance and hence controlling the resistance at this interface can have a significant impact on the electrochemical capability of the cell [46]. The real charge transfer impedance after the formation cycle comes in the order of NCM-CB < NCM-CNF < NCM-CNTs (Figure 4a). Additionally, the bulk impedance of these cells highlighted by the crossing of the Z_re_ axis is reasonably well grouped. However, the charge transfer impedance of these samples changes after 50 cycles. The charge transfer impedance is minimized in the NCM-CNF sample, with the NCM-CB and NCM-CNTs growing in impedance. The reason for the high impedance of the NCM-CB and NCM-CNTs is believed to be because of the side reaction between the sulfide SE and amorphous carbon on the CB and CNTs [49], which leads to a significant increase in the impedance of the lithium-ion and electron mobility [50]. This is also corroborated by the fact that the bulk impedance for the NCM-CB and NCM-CNTs is increased significantly, representing what is likely a breakdown of the electrolyte, reducing its ion conduction properties. Additionally, as the potential in the cathode electrode will likely be at maximum at the carbon additive surface, the surface area of the carbon additive in contact with the SE may also play an important factor in the deterioration of the electrolyte. These factors indicate that defining a suitable carbon additive for use in the SE requires a multi-faceted approach centered around morphology and crystallinity, which both contribute to the stability of the carbon additive in the SE. 

Differential capacity (dQ/dV) curves were analyzed to better understand the effect of different conductive additives on the charge and discharge properties of the NCM-CB, NCM-CNTs and NCM-CNF (Figure 5a–c). The dQ/dV curves usually display a triplet of redox peaks due to the phase transitions from hexagonal (H1) to monoclinic (M), monoclinic (M) to hexagonal (H2), and hexagonal (H2) to hexagonal (H3) during the delithiation and lithiation processes within a potential range of 1.88–3.68 (V vs. Li-In) [51]. These phase transitions are well defined through previous research and shifts in their potential are well understood to be related to the degradation of the cathode material and the cell resistance [52,53,54]. For the NCM-CNF composite, the change in potential over the 50 cycles (0.16 V vs. Li-In) is far lower than that of the NCM-CB (0.26 V vs. Li-In) and NCM-CNTs (0.25 V vs. Li-In), which can be attributed to lower internal resistance buildup within the cell [55,56]. In the case of this analysis, it is mainly the phase transition from H1 to M that was clearly displayed during cycling at 0.1 C. In the case of the NCM-CB and NCM-CNTs, the peak of the phase transition indicating H1 to M is shown at 3.21 V at 1 cycle for both, but the peak is shifted to 3.47 V and 3.46 V, respectively, after 50 cycles. However, the peak of the NCM-CNF is shown at 3.12 V at 1 cycle and 3.28 V after 50 cycles. This finding indicates that the NCM-CNF exhibited a more stable reversibility for the phase transition of the H1-M during cycling than that of the NCM-CB and NCM-CNTs. The charge–discharge capacities at each of these cycles where the dQ/dV was analyzed are included in Appendix A. 

## 4. Conclusions

In conclusion, the relationship between the electrochemical performance and the morphology and crystallinity of the carbon additive in the cathode composite was analyzed using sulfide SE-based ASSLB configurations. Three types of materials were used as conductive additives: (i) carbon black, which provides a sub-micron electron path and consists of low-crystallized carbon/amorphous carbon; (ii) CNTs, which provide sub-micron electron paths and are crystalline carbon; and (iii) CNF, which provides an electron pathway in the tens of microns and is composed of crystalline carbon. As a result of the ASSLB evaluation of these three carbon additives, the NCM-CNF shows the best electrochemical performance, resulting in a 50th cycle retention of 83.9%, with the relatively low overpotential and low polarization during the cycle and rate evaluations. On the other hand, the CNTs and CB showed high overpotential, low coulombic efficiency, and a low discharge capacity due to side reactions with the solid electrolyte. When CNF, which forms a long electron path due to the material morphology, is used as a conductive additive, it has low interfacial charge transfer impedance during the cycling and helps prevent irreversible phase changes within the cathode composite.

## Figures and Tables

**Figure 1 nanomaterials-13-03065-f001:**
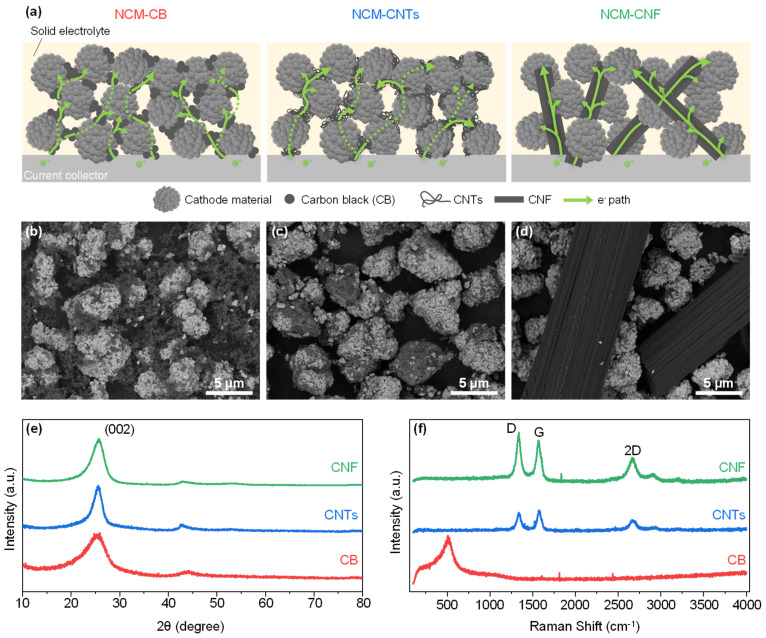
(**a**) Schematic outlining the electron pathways associated with carbon black (CB), carbon nanotubes (CNTs), and carbon nanofiber (CNF) in sulfide solid electrolyte cathode composites. Scanning electron microscopy (SEM) imaging of the surface of NCM with (**b**) Super C, (**c**) CNTs, and (**d**) CNF. (**e**) Powder X-ray diffraction (XRD) of Super C, CNTs, and CNF. (**f**) Raman spectra for Super C, CNTs, and CNF with table of associated values.

**Figure 2 nanomaterials-13-03065-f002:**
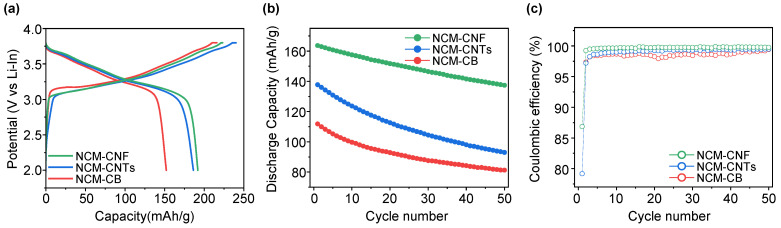
Electrochemical performance of NCM-CB, NCM-CNTs, and NCM-CNF in evaluation of all-solid-state lithium-ion batteries (ASSLBs). (**a**) Initial charge–discharge voltage profiles at 0.05 C rate between 1.88 and 3.68 V at 30 °C. (**b**) Fifty-cycle data for cathode composites applying NCM-CB, NCM-CNTs, and NCM-CNF at 0.5 C at 30 °C. (**c**) Coulombic efficiency with charge and discharge C-rates of 0.5 C and 0.5 C, respectively.

**Figure 3 nanomaterials-13-03065-f003:**
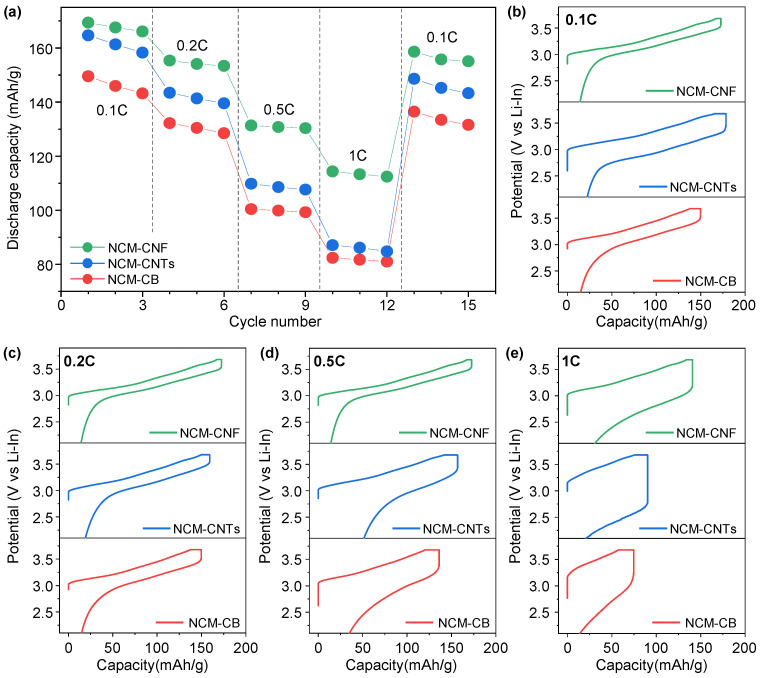
(**a**) Rate capability of NCM-CB, NCM-CNTs, and NCM-CNF at discharge current rates of 0.1 C–1 C within the potential range of 1.88–3.68 (V vs. Li-In). Voltage profile of (**b**) 0.1 C, (**c**) 0.2 C, (**d**) 0.5 C, and (**e**) 1 C showing charge and discharge cycles of NCM-CB, NCM-CNTs, and NCM-CNF cells. The curves come from the rate data in Figure 3a.

**Figure 4 nanomaterials-13-03065-f004:**
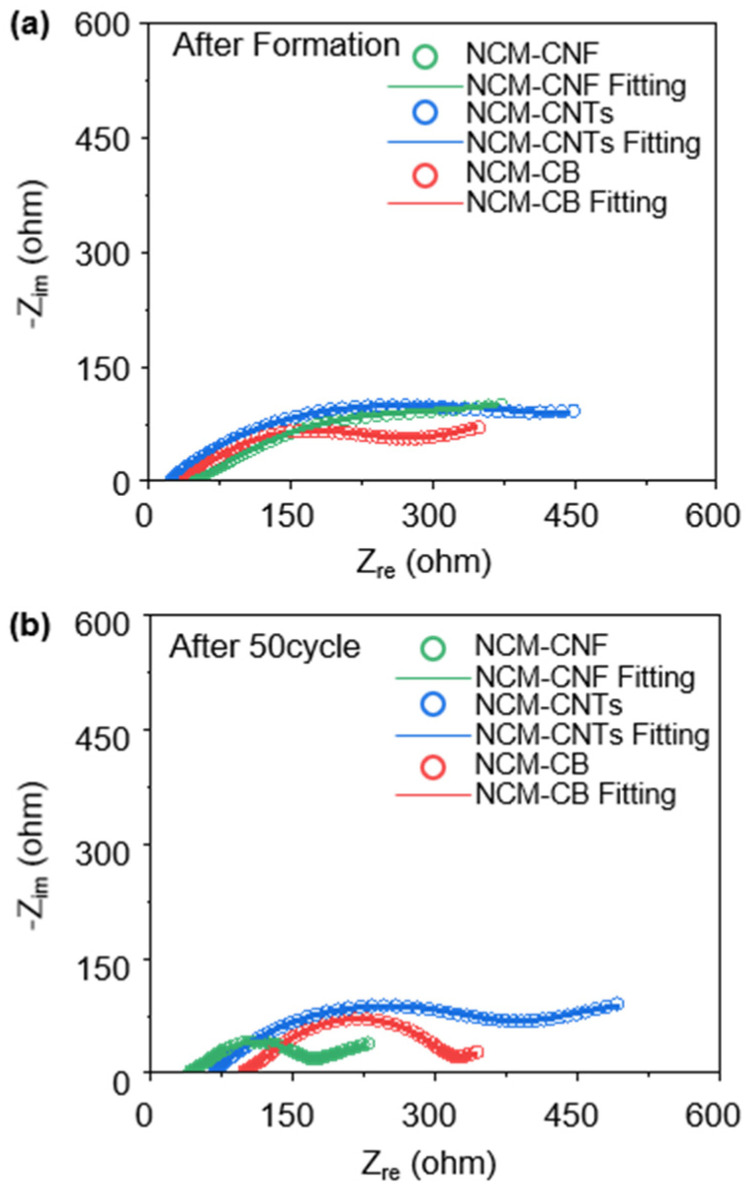
(**a**) Electrochemical Impedance spectroscopy (EIS) of cells containing NCM-CB, NCM-CNTs, and NCM-CNF taken (**a**) after the formation cycle at 0.05 C and (**b**) after 50 cycles at 0.5 C.

**Figure 5 nanomaterials-13-03065-f005:**
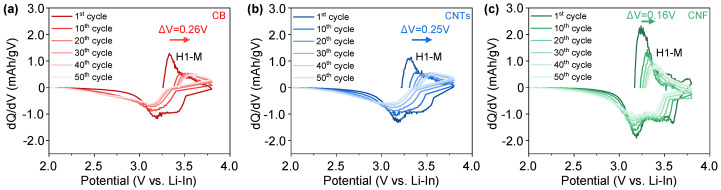
dQ/dV analysis for the cycle charge/discharge of cells containing (**a**) NCM-CB, (**b**) NCM-CNTs, and (**c**) NCM-CNF at 0.5 C.

**Table 1 nanomaterials-13-03065-t001:** D band, G band, and value of I_D_/I_G_ of NCM-CB, NCM-CNTs, and NCM-CNF.

	D Band (cm^−1^)	G Band (cm^−1^)	Intensity Ratio (I_D_/I_G_)
NCM-CNF	1339.1	1566.4	1.037
NCM-CNTs	1342.6	1574.8	0.989
NCM-CB	---	---	---

**Table 2 nanomaterials-13-03065-t002:** Charge capacity, discharge capacity, and initial coulombic efficiency (ICE) of NCM-CB, NCM-CNTs, and NCM-CNF at 0.05 C at 30 °C. Charge and discharge capacity of 1 cycle, 50 cycles, and capacity retention after 50 cycles at 0.5 C and 0.5 C.

	Charge Capacity (mAh/g)	Discharge Capacity (mAh/g)	ICE (%)	Discharge Capacity (mAh/g) @1cycle	Discharge Capacity (mAh/g) @50cycle	Discharge Retention (%) @50cycle
NCM-CNF	223.04	191.78	85.98	163.6	137.4	83.9
NCM-CNTs	240.00	186.18	77.58	137.8	93.0	67.4
NCM-CB	215.77	152.01	70.45	111.8	81.2	72.6

## Data Availability

Data is contained within the article and Appendix A.

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
