# Peer review of "The Effect of Conductive Additive Morphology and Crystallinity on the Electrochemical Performance of Ni-Rich Cathodes for Sulfide All-Solid-State Lithium-Ion Batteries"

_nanomaterials, 2023, doi:10.3390/nano13233065_

Round 1

Reviewer 1 Report

Comments and Suggestions for Authors

The rapid development of electrical vehicles and mobile gadgets has motivated the growing demand for batteries with high energy density. The organic solvent-based liquid electrolytes are flammable, therefore, the authors in the current work have chosen to solve these issues associated with the liquid electrolytes and replace them with solid-state electrolytes. The sulfide-based solid-state electrolytes (maybe) have drawn a lot of attention because of their high Li-ion conductivities, and easy processability, however, the authors must address this in the appropriate section. The manuscript is OK, and there is some novelty. However, the work requires major revision before rendering any final decision.

My specific points are below:

·         The title is very confusing. It should be something along these lines “The role of conductive carbon with their effect on morphology and crystallinity…”

·         Abstract – how effective are the carbon black versus carbon nanotubes in improving ionic conductivity?

·         Are these two additives also tested as combined?

·         What is the optimized value of these additives in the electrode?

·         What is the rationale for choosing sulfide electrolyte? What do you call “Li6PS5Cl SSEs” is there any structure?

·         Introduction – Please provide the theoretical voltage and energy density of the LIB.

·         What are the common cathodes used in LIBs? Just add one or two sentences

·         Page 1, lines 44 – not all liquid electrolytes are flammable. The aqueous electrolytes reported by M. Minakshi et al (doi.org/10.1016/j.jpowsour.2004.06.049) are safe and affordable. Discuss.

·         Line 84, what does SE stand for?

·         Line 99, usually called as NMC.

·         Lines 69 – 72 The additives like alkaline earth oxides (BaO) doi.org/10.1016/j.jallcom.2011.03.044 have also been reported in the domain with the improved energy density. Please discuss.

·         The cycling performance of the ASSLBs with Li6PS5Cl electrolyte is usually poor, how the authors mitigate this? How this electrolyte has been prepared?

·         Is there any fitting curve for EIS plots?

·         From Figure 5, please quantify the electrochemical performance by providing the cell capacity in mAh/g.

Comments on the Quality of English Language

The overall language must be improved.

Reviewer 2 Report

Comments and Suggestions for Authors

Recommendation: major revision.

Comments: This manuscript investigated the impacts of carbon black (CB), carbon nanotubes (CNTs), and carbon nanofiber (CNF) on the conductivity, electrochemical performance of LiNi0.8Co0.1Mn0.1O2 (NCM811)-based solid-state batteries (SSBs). The cycling performance and rate capability of these SSBs are investigated. Generally, the manuscript provides sufficient research inductively by experimental results, which is convenient for readers to learn about the effect of carbon material on the electrochemical performance of NCM811 SSBs. I recommend that this paper maybe published in Batteries after a major revision. Some small issues that should be solved are the following:

1. The introduction of this paper needs to make a strong argument about the impact and novelty of the work. So, the introduction should enrich some related articles in this section. Such as ACS Energy Lett. 2023, 8, 4903; Adv. Energy Mater., 2021, 11, 2003583.

2. The better identify these carbon materials states in cathode composites, high-resolution SEM or TEM images are better offered.

3. The active mass loading of cathode and electrolytes is 15 mg and 150 mg, respectively. The energy density of these SSBs is too low for practical application.

4. The authors emphasized the dQ/dV(Figure 5) curves display a triplet of redox peaks due to the phase transitions from hexagonal (H1) to monoclinic (M), monoclinic (M) to hexagonal (H2), and hexagonal (H2) to hexagonal (H3) during delithiation and lithiation processes. These phases of NCM811 should be confirmed by XRD.

5. The authors better compared their electrochemical performance with reported NCM811 SSBs

6. Some formatting issues in the References section, the authors should carefully check and correct them.

Reviewer 3 Report

Comments and Suggestions for Authors

I have read through the manuscript titled “Analysis of Electrochemical Performance According to Morphology and Crystallinity of Conductive Additives for Sulfide All-Solid-State Lithium-ion batteries”. In this manuscript, the relationship between the electrochemical performance of SE-based ASSLB with the morphology and crystallinity of the carbon additive in the cathode mixture was analyzed in detail. the NCM-CNF shows the best electrochemical performance with the relatively low overpotential and low polarization. This is a very detailed work, and can provide reference value for the preparation of high-performance all-solid-state batteries. However, there are many details in the manuscript that need to be revised. Therefore, it is recommended to be published in the journal of nanomaterials after a major revision.

1.     Due to the repeated use of many words and simple sentence structure, this manuscript needed further polishing (In the case of….., On the other hand……) .

2.     Table 1 display the ID/G value, not the electrochemical data. There is an error in the description in line 212. Moreover, there is no G-peak and 2D-peak for CB in Figure 1f, how was the ID/G value in Table 1 calculated?

3.     The manuscript described the overpotential of CB higher than the other two conductors in line 212, please calculate the value of the overpotential according to the results in Figure 2.

4.     Does ΔV have any special significance in Figure 5? Please explain this value in detail. In addition, how to reflect the reversibility for the phase transition of H1-M by the results of differential capacity (dQ/dV)? It is suggested to analyze the data in detail or cite relevant literature.

5.     To further enrich the background introduction and discussion, some latest research studies about LIBs should be cited to deepen the discussion, such as 10.1002/adma.202307091; Angew. Chem. Int. Ed., 2023, 62, e202218672; etc.

Comments on the Quality of English Language

Minor editing of English language required

Round 2

Reviewer 1 Report

Comments and Suggestions for Authors

The authors have addressed my queries/concerns raised earlier. Therefore, in this reviewer's opinion, the revised manuscript is suitable for publication.

Reviewer 2 Report

Comments and Suggestions for Authors

I am satisfied with the modification made to this manuscript, which can be accepted now.